# Atherosclerosis and the Bidirectional Relationship between Cancer and Cardiovascular Disease: From Bench to Bedside—Part 1

**DOI:** 10.3390/ijms25084232

**Published:** 2024-04-11

**Authors:** Giuseppina Gallucci, Fabio Maria Turazza, Alessandro Inno, Maria Laura Canale, Nicola Silvestris, Roberto Farì, Alessandro Navazio, Carmine Pinto, Luigi Tarantini

**Affiliations:** 1Independent Researcher, 85025 Melfi, Italy; pina.gallucci@tiscali.it; 2Struttura Complessa di Cardiologia, Fondazione IRCCS Istituto Nazionale dei Tumori, 20133 Milano, Italy; turazzafabio@gmail.com; 3Oncologia Medica, IRCCS Ospedale Sacro Cuore Don Calabria, 37024 Negrar di Valpolicella, Italy; alessandro.inno@sacrocuore.it; 4Division of Cardiology, Azienda USL Toscana Nord-Ovest, Versilia Hospital, 55041 Lido di Camaiore, Italy; marialaura.canale@uslnordovest.toscana.it; 5Medical Oncology Unit, Department of Human Pathology “G.Barresi”, University of Messina, 98100 Messina, Italy; silvestrisnicola@gmail.com; 6Clinical and Experimental Medicine PhD Program, University of Modena and Reggio Emilia, 41100 Modena, Italy; 7Cardiologia Ospedaliera, Department of Specialized Medicine, AUSL—IRCCS in Tecnologie Avanzate e Modelli Assistenziali in Oncologia, 42100 Reggio Emilia, Italy; alessandro.navazio@ausl.re.it; 8Provincial Medical Oncology, Department of Oncology and Advanced Technologies, AUSL—IRCCS in Tecnologie Avanzate e Modelli Assistenziali in Oncologia, 42100 Reggio Emilia, Italy; carmine.pinto@ausl.re.it

**Keywords:** atherosclerosis, cardiovascular disease (CVD), cancer, inflammation, immune system, endothelium, exposome

## Abstract

Atherosclerosis, a complex metabolic-immune disease characterized by chronic inflammation driven by the buildup of lipid-rich plaques within arterial walls, has emerged as a pivotal factor in the intricate interplay between cancer and cardiovascular disease. This bidirectional relationship, marked by shared risk factors and pathophysiological mechanisms, underscores the need for a comprehensive understanding of how these two formidable health challenges intersect and influence each other. Cancer and its treatments can contribute to the progression of atherosclerosis, while atherosclerosis, with its inflammatory microenvironment, can exert profound effects on cancer development and outcomes. Both cancer and cardiovascular disease involve intricate interactions between general and personal exposomes. In this review, we aim to summarize the state of the art of translational data and try to show how oncologic studies on cardiotoxicity can broaden our knowledge of crucial pathways in cardiovascular biology and exert a positive impact on precision cardiology and cardio-oncology.

## 1. Introduction

Cardiovascular disease (CVD) and cancer are the leading causes of morbidity and mortality worldwide [1]. CVD, the most common noncommunicable condition, is responsible for almost one third of all deaths globally [2]. In Europe, CVD is the most common cause of death, with more than 60 million potential years of life lost annually. Mortality is higher in women compared to men, while age-standardized rates of morbidity and death from CVD are higher in men than in women [3]. In 2020, it was estimated that 19.3 million new cancer cases and almost 10.0 million cancer deaths occurred worldwide. Female breast cancer (BC) was the most commonly diagnosed cancer in 2020, with an estimated 2.3 million new cases and lung cancer was the leading cause of cancer death, with an estimated 1.8 million deaths [4]. The global cancer burden is increasing and is expected to be 28.4 million cases in 2040, a 47% rise from 2020. Recent epidemiological evidence underscores that CVD and cancer are supported by shared risk factors and that 40% of cancer cases are indeed promoted by environmental factors, the most important of which are well-known manageable cardiovascular risk factors (CVRFs) such as smoking, consuming alcohol, and an unhealthy lifestyle linked to obesity [5]. Furthermore, a high-risk score for atherosclerotic cardiovascular disease (ASCVD) predicts the future development of neoplasms [6], as well as the presence of coronary calcifications in computed tomography (CT) scans [7]. The sharing of the same risk factors between cancer and ASCVD suggests the existence of common pathogenetic mechanisms in which chronic inflammation plays a major role [8]. Among patients receiving contemporary statins therapy, inflammation assessed by high-sensitivity C-reactive protein (hs-CRP), a marker of an activated innate immune system, is a stronger predictor for risk of future cardiovascular events and death than cholesterol assessed by low-density lipoprotein-Cholesterol (LDL-C) [9,10]. These observations in a large number of patients support the results of clinical trials with anti-inflammatory agents such as monoclonal antibodies against interleukin (IL)-1β, IL-6, and colchicine in patients with ASCVD [11,12,13,14,15]. Unexpected data for canakinumab included the lower incidence of lung cancer [16]. This preliminary observation is coupled with the reduced incidence of cancer in gout patients treated with colchicine, indicating the importance of inflammation and innate immunity in the development of tumors [17]. In addition, recent data indicate a potentially significant role of the inflammatory state in predicting prognosis and response to immunotherapy in cancer patients [18,19]. Increasing real-world evidence indicates that vascular damage and related ASCVD can pose a significant problem for the outcome of current cancer treatments.

The aim of this review is to summarize translational and clinical data on ASCVD and cancer and the impact of these data on precision cardiology and cardio-oncology, with a focus on preventive strategies.

## 2. The Changing Paradigm of Atherosclerosis and Related Cardiovascular Disease

Atherosclerosis is a chronic vascular disease characterized by the presence of fibroinflammatory lipid plaques in large- and medium-sized arteries. Although clinical ASCVD usually affects the elderly, it is well known that the process begins in utero and evolves silently throughout life, depending on genetic predisposition and, to a greater extent, by acquired factors due to certain behaviors and exposure to environmental factors. Over decades, a lot of research in clinical medicine, pathology, epidemiology, and public health has led to the identification and characterization of environmental factors, biological agents, disease conditions, and genetic states that increase the likelihood of atherosclerosis, the main cause of CVD. To encourage an inclusive treatment paradigm for promoting and preserving cardiovascular health throughout life in populations and individuals, in 2010 the American Heart Association (AHA) defined “healthy diet, exercise, nicotine avoidance, healthy weight, healthy values of blood pressure, blood lipids, blood glucose” as “Life’s simple 7”: seven parameters to maintain CV health [20]. In 2022, the AHA “Life’s Simple 7” was upgraded to “Life’s Essential 8” adding “healthy sleep” as an essential component of CV health and emphasizing the crucial role of psychological and sociological issues [21]. There is indeed an increasing interest in the link between psychological health/well-being/social determinants of health and CV health due to the favorable effect of factors such as optimism, purpose in life, and resilient coping and the negative effect of psychosocial stress and depression [22,23]. Moreover, some chronic stress-related pathways such as inflammatory response, glucose and lipid homeostasis, and coagulation have been identified [24]. At the same time, a growing interest in the concept of “exposome” is arising in the field of preventive cardiology to underline the importance of non-traditional risk factors in the development/progression of atherosclerosis [25,26] and in connected clinical chronic conditions (e.g., diabetes) [27]. The term “exposome” was coined in 2005 by Wild [28] to weigh the impact of exposure to environmental agents on molecular pathways (genetic configuration) that, by changing internal homeostasis, lead to chronic diseases. The exposome is an aggregation of the general environmental exposome (air, soil, light, radiation, chemical and noise pollution) and the personal exposome related to individual/lifestyle factors (exercise, diet, infection, drug, stress, sleep deprivation, intestinal flora) [29]. The reasons for such interest are certified by 2021 Global Disease Burden data where unhealthy “dietary risk” and air pollution have passed cigarette smoking in the ranking of CV risk factors [30] and by the newest evidence of the clinical impact of microplastic and nano-plastic contaminants in atherosclerotic plaques [31,32]. Almost all the components of the exposome raise the endocrine-sympathetic response to stress, alter the metabolic state in the obesogenic sense, and increase oxidative stress and inflammation [33,34,35]. Furthermore, they affect hematopoiesis, the circulating leukocytes, and the recruitment of monocytes in the atherosclerotic plaque [36]. In the final analysis, the trajectory of CVD is increasingly recognized as being shaped by intricate interactions between polygenic factors and environmental influences. However, unlike blood pressure, blood sugar, or blood lipid levels, practical and universally recognized tools are currently lacking to incorporate the impacts of environmental (such as pollution), socio-economic (including social deprivation, limited access to a healthy diet, opportunities for regular physical activity, and suboptimal healthcare), and psychological factors (such as stress, anxiety, and depression) into individual patient management [21,37]. As precision biomedicine advances and genomic science expands our understanding of biological systems and molecular networks underlying CVD, it is likely that we will develop tools to address these complexities. Novel methods, such as immune phenotyping and “multi-omic” characterization of the epigenome, transcriptome, metabolome, and microbiome hold promise for accelerating the discovery of biomarkers and identifying individuals at higher risk. Currently, we can only consider their presence as “modifiers” of the risk ascertained by the classic parameters and modulate accordingly, if possible, the treatment plan for the individual patient. What we observe clinically, indeed, is that over time, the landscape of atherosclerosis has changed [38]. A key factor is the considerable increase in obesity, usually associated with hypertension, and related metabolic changes such as insulin resistance, metabolic syndrome, type 2 diabetes, and changes in lipid patterns characterized by low high-density lipoprotein (HDL) cholesterol and elevated triglyceride-rich lipoprotein [39,40,41,42,43,44]. Nowadays, traditional CVRFs may fail to predict CVD in the individual patient. In the PESA (Progression of Early Subclinical Atherosclerosis) study, an ongoing longitudinal cohort study “integrating serial imaging, biological and behavioral parameters associated with the progression of subclinical atherosclerosis, 63% of the 4184 enrolled asymptomatic subjects (average age 46 years) had, in the basal examination, atherosclerosis in one or more vascular districts (iliofemoral, carotid, coronary), and a third of them had a low risk according to traditional risk estimates (Framingham Heart Score < 10%) [45]. In addition to confirming the concept that “lower is better” for cholesterol, the PESA study highlighted the role of triglycerides (TG) in atherosclerosis and vascular inflammation in low- and moderate-risk patients [46], and this result was recently confirmed in other registries [47,48]. In clinical practice, one of the most frequent causes of hypertriglyceridemia is associated with insulin resistance in type 2 diabetes mellitus. High levels of TG are a component of metabolic syndrome and are associated with the development of atherosclerosis in non-alcoholic fatty liver disease (NAFLD), where TG and remnants play a significant role. The growth of atherosclerotic plaque is a chronic dynamic inflammatory process with many different factors. Early in atherogenesis, high levels of oxidized low-density lipoproteins (ox-LDL) and remnants of triglyceride-rich lipoproteins gain access to the subendothelial space, eliciting a danger signal that activates the NOD [nucleotide oligomerization domain]-containing, LRR [leucine-rich repeat]-containing, and PYD [pyrin domain]-containing protein3 (NLRP3) inflammasome [49,50] in innate immune cells. The activated inflammasome starts a cascade of inflammatory cytokines (IL-1β, IL-6) and upregulates high-sensitivity C-reactive protein (hs-CRP) in the liver; this eventually leads to atherosclerotic lesions. These processes disrupt the integrity of endothelial cells (ECs) and their atheroprotective role, which includes secretion of many vasoactive substances affecting vasodilatation, platelet function, and monocyte infiltration. As a consequence, the activated “inflamed” ECs steer the recruited immune cells towards the adoption of their proinflammatory phenotypes, expanding the process of tissue inflammation [51,52]. Prolonged stimulation of ECs may also contribute to the endothelial-to-mesenchymal transition that leads to fibrosis [53]. The recruitment of immune cells has a pivotal role in atherosclerosis and seems to be the target to treat residual CV risk (beyond the lipid lowering therapies). For this reason, monocytes, macrophages, dendritic cells, neutrophils, T cells, and B cells are all deeply involved in the inflammatory scenario [54]. Their behavior (how they move and change shape) comes from genetic and signaling networks [55]. In recent decades, a growing interest in microbial stimuli-induced and endogenous ligand-induced upgrading of innate immune cells that allows an increased response against a secondary stimulation (the so-called “trained immunity”) has been widely documented. Unfortunately, “trained immunity” may also have a negative effect. When activated in an inappropriate way, it can lead to inflammatory and autoimmune diseases; trained immunity, for example, confers a proatherogenic phenotype to monocytes and macrophages [56,57,58]. In conclusion, there has been a paradigm shift in atherogenesis: the historical view of a single culprit agent inducing atherosclerosis (either lipoproteins [59] or inflammation [60]) has been upgraded to a complex metabolic-immune process in which a metabolic switch of ECs starts the inflammatory process that involves many immune and vascular cells. Moreover, the complexity of the challenges to ECs has also expanded in recent decades, and noxious substances or abnormal hemodynamic stresses are assembled in the exposome concept [25]. The intriguing chameleonic activity of vascular cells (ECs [51,52,58,61], pericytes [62,63,64,65], smooth muscle cells [66,67,68,69]) and immune cells (dendritic cells [70], monocytes [71,72], macrophages [73,74,75,76,77], T cells [78,79,80], B cells [81,82,83], polymorphonuclear neutrophils (PMNs) [84,85,86]) that adapt their phenotypes to the multifaceted scenario of atherosclerosis is summarized in Table 1.

The remarkable inflammatory features of atherosclerosis explain the beneficial effect of drugs that target the inflammasome pathway, but a “focused” cytokine inhibition is necessary, as Ridker states, to yield an effective atheroprotective effect [87,88]. Some of the trials on the atheroprotective effects of anti-inflammatory agents are summarized in Table 2.

In the conundrum of atherosclerosis, there are risk-enhancing factors such as Lipoprotein(a) [Lp(a)] [89,90] and the microbiome [91,92]. Lp(a) seems to be the strongest independent genetic risk factor for myocardial infarction and aortic stenosis [93] and is associated with increased mortality [94]. There are also somatic mutations in hematopoietic stem cells, with subsequent clonal expansion of hematopoietic cells. Clonal hematopoiesis of indeterminate potential (CHIP) is associated with an expected 0.5–1.0% risk per year of leukemia but with an unexpected two-fold increase in cardiovascular risk independent of traditional risk factors. CHIP has to be considered a new CVRF [95,96,97] and plays an important role in increasing CV complication in cancer patients [98]. In recent years, there has been a growing interest in CHIP due to TET-2 loss of function variants because preclinical studies have suggested that Tet2 loss of function in myeloid cells, and the subsequent increase in IL-1 β-related signaling, act as accelerators of atherosclerosis [99]. These observations promoted the hypothesis that IL-1β neutralization would lead to a greater reduction in MACEs, and this was indeed the case in a subgroup analysis of the CANTOS trial that included patients with a history of myocardial infarction, elevated high-sensitivity C-reactive protein, and CHIP variant *TET2* clones [100].

**Table 2 ijms-25-04232-t002:** Anti-inflammatory agents and CVD.

Study	Anti-Inflammatory Agent	Target	Population	Effect on Inflammation Biomarker	Clinical Effects	References
CANTOS (main study) subgroup analysis of CANTOS	3 doses of canakinumab (50 mg, 150 mg, and 300 mg) subcutaneously (s.c.) every 3 months vs. placebo	IL-1β	10,061 patients with previous myocardial infarction and hsCRP ≥ 2 mg/L Subgroup of 338 patients with clonal haematopoiesis and variants in *TET2* more common than *DNMT3A*	Reduction in hsCRP (for all the doses)	The dose of 150 mg s.c. every 3 months was associated with a significant reduction in recurrent CV events Patients with CHIP due to somatic variants in *TET2* had reduced risk for MACE	Ref. [11] NEJM 2017;377:1119 Ref. [100] JAMA Cardiol 2022;7:521
CIRT	Low-dose methotrexate (15–20 mg weekly)	No specific target	4786 patients with known atherosclerosis and either DM orMS	No reduction in IL-1β or IL-6	No reduction in CV event rates	Ref. [87] NEJM 2019;380:752
RESCUE	Ziltivekimab (7·5 mg, 15 mg, or 30 mg every 4 weeks up to 24 weeks)	IL-6	264 patients with high CV risk> (age ≥ 18 years, moderate to severe CKD, hsCRP ≥ 2 mg/L)	Reduction in biomarkers of inflammation (hsCRP) and thrombosis (e.g., fibrinogen)	Reduction in biomarkers of inflammation (hsCRP) and thrombosis (e.g., fibrinogen)	Ref. [12] Lancet 2021;397:2060
COLCOT	Low-dose colchicine (0.5 mg daily)	Inhibition of tubulin polymerization and alteration in leukocyte responsiveness	Patients with recent myocardial infarction: 2366 patients assigned to colchicine and 2379 to placebo		Significant reduction in ischemic CV events	Ref. [13] NEJM 2019;381:2497
Lo-Do-Co2	Low-dose colchicine (0.5 mg daily)	Inhibition of tubulin polymerization and alteration in leukocyte responsiveness	Patients with chronic coronary artery disease in stable condition: 2762 patients assigned to colchicine and 2760 to placebo.		31% lower relative risk of CV death, spontaneous myocardial infarction, ischemic stroke, or coronary revascularization in patients treated with colchicine compared to placebo	Ref. [14] NEJM 2020;383:1838

CANTOS, Canakinumab Anti-Inflammatory Thrombosis Outcomes Study; CIRT, Cardiovascular Inflammation Reduction Trial; COLCOT, Colchicine Cardiovascular Outcomes Trial; CV, cardiovascular; hsCRP, high-sensitivity C-reactive protein.

Another interesting component of atherosclerotic cardiometabolic derangements is epicardial adipose tissue (EAT), which can be both useful for the myocardium (offering a thermogenic function) and dangerous (through its paracrine or vasocrine secretion of proinflammatory and profibrotic cytokines). EAT changes with age and in pathologic conditions and is considered proatherogenic for coronary arteries [101,102].

The complexity of the inflammation/immunity process in acute myocardial infarction (AMI) has been recently translated into a play on words of the well-known Western film as “Good”, “Bad”, and “Ugly” characters. The “Good” players are T cells, natural killer cells (NKs), and macrophages that protect and heal the myocardium along with the good cytokines (IL-10 and IL-2) that decrease proinflammatory signals [tumor necrosis factor α (TNFα), monocyte chemoattractant protein-1 (MCP-1), IL-8], reduce extracellular matrix remodeling, and promote the activation of regulatory T cells (T reg), type 2 helper T cells (Th2), and NKs, promoting the protective M2 phenotype of macrophages. The “Bad” players are the M1 macrophages, the foam cells, and PMNs; together, they maintain a low-grade inflammation status in the late phase after AMI, inducing the NLRP3 inflammasome and increasing the production of the bad IL-1α, IL-1β. The “Ugly” players are the activated PMNs, the recruited monocytes, IL-1, and IL-6; they act in the early phase after AMI soon after plaque rupture and thrombosis. This process may be amplified by NET formation [103]. Some of these characters may also be involved in the process of atherosclerosis, as shown in Table 1 and in the central illustration.

## 3. Atherosclerosis and Cancer: The Unexpected Link

Over the years, community studies have documented that adherence to the seven ideal health metrics defined in the AHA goals [20] is associated with lower cancer incidence [6,104,105,106,107]. Cancer and CVD are intertwined [108,109,110,111] by the sharing of the same risk factors (see Koene [112] for more details) and of the fundamental physio-pathological mechanism that is represented by chronic inflammation.

## 4. Atherosclerosis and Cardio-Oncology: The Bidirectional Relationship Has Highlighted the Novel Issues That Need to Be Addressed

### 4.1. The Shared Risk Factors

Obesity, hypertension, diabetes, smoking, dyslipidemia, physical inactivity and sedentary behavior, unhealthy diets, alcohol abuse, impaired immune response, metabolic remodeling, and CHIP are risk factors for both CVD and cancer and represent the different epiphenomena of the common characteristics that underline the two most frequent noncommunicable diseases [113].

### 4.2. The Shared Pathways

The hallmarks of cancer (sustained proliferation, resistance to cell death, neurohormonal stress, genomic instability, and inflammation) [114] also have a pivotal role in CVD. The following biological processes shared by cancer and CVD and the candidate biomarkers that represent these shared biological processes are illustrated according to a seminal review by Narayan et coll [115].

-Inflammation, an epiphenomenon of immune dysregulation and cell senescence, is associated with increased levels of hsCRP and IL-6 and suppression of tumorigenicity 2 (ST2). These biomarkers are linked to tissue invasion and metastasis associated with cancer but also with tissue damage that underlies the atherosclerotic process [116,117].-Cellular proliferation, a mitogenic function, whose marker is Galectin-3, can stimulate not only proliferation (through paracrine interactions) but also cardiovascular remodeling and cardiac fibrosis [118,119].-Resistance to cell death is linked to cellular stress response and apoptosis; its biomarker, growth differentiation factor (GDF-15), has a prognostic role in cancer mortality and in CVD (myocardial infarction, thromboembolic stroke, heart failure, and stroke) [120,121]; cardiac Troponin T (cTnT), a well-known marker of myocardial cell death, is also a useful marker in light chain amyloidosis [122,123].-Neurohormonal stress leads to increased levels of cardiovascular neurohormones such as N-terminal pro-B-type natriuretic peptide, mid-regional pro-atrial natriuretic peptide and other neurohormones, and diuretic hormones. These cardiovascular neurohormones have a relevant role in patients with acute or chronic heart failure (HF), but they might also play a role in cancer, since they may be produced by some malignant cells in the vascular bed of tumors [124,125].-Angiogenesis, with a role in EC survival and in tumorigenesis, invasion, and metastasis, may be measured by angiogenic biomarkers such as soluble fms-like tyrosine kinase 1, a variant of Flt-1 known also as vascular endothelial growth factor (VEGF) receptor, and placental growth factor (PlGF). VEGF biology has a relevant impact on tumorigenesis and on normal cardiovascular function [126,127].-Genomic instability, such as CHIP, has an impact on both CVD and cancer. CHIP [128] is a risk factor for CVD [95], but it may be caused by atherosclerosis due to the continuous stimulation of stem cell proliferation [129]. This reverse CHIP effect has uncovered an unexpected link between oncogenesis and atherogenesis [38]. The biological explanation of the impact of a potentially precancerous lesion on CVD involves an interaction between clonally derived monocytes and macrophages and the vascular endothelium that leads to vascular inflammation and accelerated atherogenesis [95,128].

The shared pathways are illustrated in the central illustration (Figure 1).

### 4.3. The Atherogenic Effect of Some Oncologic Treatments

-Radiotherapy-induced endothelial dysfunction. Radiation-associated damage induces the secretion of proinflammatory cytokines; increases the release of reactive oxygen species (ROS); causes a dysregulation of glycolysis, lipid metabolic pathways, and angiogenesis; and may have a negative impact on telomere function and immunity homeostasis. The final effect is EC death (either acutely via apoptosis or chronically via EC senescence) and a disrupted EC environment [130]. Clinical phenotypes of radiation-associated vascular damage are accelerated CAD, cerebral events due to carotid artery disease, calcification of the ascending aorta and aortic arch, and lesions of other vascular segments in the radiation field [131]. The recent BACCARAT study evaluated the association between cardiac exposure and the risk of developing calcified and non-calcified atheromatous plaques within 2 years of RT. As both calcified and non-calcified plaques were found, it may be hypothesized that cardiac radiation exposure accelerates the process of atherosclerosis in already existing plaques with an increase in their calcium content (calcified plaques) and starts new non-calcified plaques [132].-Cancer therapy-induced vasculotoxicity is associated with traditional chemotherapies (alkylating agents, microtubule inhibitors, and antimetabolites), with targeted therapies such as VEGF inhibitors, with breakpoint cluster region–Abelson oncogene locus tyrosine kinase inhibitors, and with multiple myeloma drugs. The majority of these drugs induce hypertension that may eventually drive atherosclerosis. They may also produce CV injury due to damage-associated molecular patterns (DAMPs) that sustain inflammation. There are many clinical phenotypes of vasculotoxicity, including CAD, stroke, systemic and pulmonary hypertension, vasospasm, and thrombosis [133,134].-Accelerated atherosclerosis is induced by immune checkpoint inhibitor (ICI) treatment. Oncologic studies of ICI-induced cardiotoxicity have indeed shed light on the complex relationship between the immune system, inflammation, and atherosclerosis. Preclinical studies have shown that the targets of ICIs [PD-1 (programmed cell death protein 1), PD-L1 (programmed death ligand 1), CTLA-4 (cytotoxic T-lymphocyte–associated protein4)] are proteins with a negative regulatory role in atherosclerosis [135]. Blockage of the checkpoints may predictably lead to accelerated atherosclerosis through enhanced T cell responses, limited T_reg_ function, and infiltration of the vascular endothelium [136,137,138,139]. Preclinical studies have also shown that short-term ICI treatment promotes DAMPs and proinflammatory cytokine production [140]. In a clinical setting, a seminal study of 2842 patients and 2842 controls matched by age, a history of cardiovascular events and cancer type showed a 3-fold increase in atherosclerotic cardiovascular events (myocardial infarction, coronary revascularization, and ischemic stroke) after starting ICI treatment. Moreover, in a case-crossover analysis performed by the same authors and comparing an at-risk period (defined as the 2-year period after ICIs) and a control period (defined as the 2-year period before ICIs), atherosclerotic cardiovascular events significantly increased from 1.37 to 6.55 per 100 person-years at 2 years; in a subgroup of 40 patients, a 3-fold-higher rate of aortic plaque progression was also documented [141]. In a more recent retrospective cohort study on 1458 patients diagnosed with stage III or IV non-small-cell lung cancer (NSCLC) treated with (487 patients) and without (971 patients) ICI therapy and followed-up for a median time of 23.1 months, ICI therapy was associated with a 3.6-fold increase in the total risk of ASCVD events before propensity score matching [142].-Hormone therapy-associated risk of dyslipidemia, atherosclerosis, and heart failure. This effect has been proven in BC women treated with aromatase inhibitors that increase the risk of atherosclerosis, HF, and dyslipidemia [143] and in prostate cancer patients treated with androgen deprivation therapy (ADT) in which the increased risk of CV events is linked to indirect modifications of CVRFs. More specifically, Gonadotropin-releasing hormone (GnRH) agonists increase LDL-cholesterol and triglyceride levels, visceral fat, and insulin resistance and decrease lean body mass and glucose tolerance, leading to accelerated atherosclerosis and coronary artery disease (CAD) events, HF, and arrhythmias [144,145]. In preclinical studies, orchiectomy and GnRH agonists, but not GnRH antagonists, induced long- or short-term follicular stimulating hormone elevation that, acting synergistically with TNF-α, induced an amplified endothelial inflammation through elevation of vascular cell adhesion protein-1 expression, thus accelerating atherosclerosis [146].

## 5. How Do We Measure and Quantify Atherosclerosis?

ASCVD can be imagined as a kind of continuum that originates from the earliest onset of vascular atherosclerosis at the cellular level, passes through a lasting, clinically silent, histological evolution, and eventually ends with overt clinical complications. The time frame within which these processes are included can reach some decades, with long phases of quiescence alternating with more or less prolonged phases of instability. The ability to intercept, in the individual patient, the evolution of atherosclerosis before it becomes clinically manifest and, thus, given the wide time frame, to ultimately prevent its most dreaded complications, has always been and still is, one of the holy grails of CV medicine. This long path, which started from the epidemiological studies that since the 1950s have enabled the identification of the best-known CVRFs [147,148,149], has recently been enriched with more refined tools capable of identifying the factors that, at the pathophysiological level, initiate the atherosclerotic process and promote the transition from stable phases to acute events; and with increasingly sensitive and specific biomarkers of atherosclerosis. These tools, at the individual level, help identify the patients who are at higher risk of acute clinical events. This goal is achieved thanks to diagnostic methods capable of investigating plaque composition and atherosclerotic lesions at higher risk of acute complications.

### 5.1. Risk Scores and Mendelian Randomization

Randomized controlled trials (RCTs) are considered the gold standard design to infer causality, and this is true even for CVRF’s role in atherosclerosis development. However, RCTs are expensive, time-consuming, and often difficult to conduct, particularly if, as is the case for CVRF identification, a long follow-up is needed. Other limits are represented by possible poor long-term compliance and ethical issues about random treatment allocation. Therefore, the relationship of modifiable CVRFs with CV events has mostly been investigated using an observational study design, e.g., through case-control studies and cohort studies. The limitations of this type of approach are the presence of confounding factors and the so-called reverse causation bias; these make any causal relationship that may be demonstrated less reliable, given that confounding factors and reverse causation bias can distort the findings. Observational and cohort studies conducted over the past 75 years, starting with the Framingham study, have enabled the construction of a number of risk scores that can be applied in different populations [150,151].

The most recent European guidelines on cardiovascular prevention in clinical practice [37] recommend the use of the Systematic Coronary Risk Evaluation (SCORE2) and Systematic Coronary Risk Evaluation-Older Persons (SCORE2-OP) systems. The risk cards in this system allow estimation of the 10-year risk of CVD in four European geographical risk regions. In Europe, maps have been developed for low-, moderate-, high-, and very high-risk regions. This rating system takes into account risk factors such as non-HDL cholesterol, systolic blood pressure, smoking, sex, and age, thus defining different risk categories (low, moderate, high, and very high) according to possibly associated additional CVRFs [152]. In recent years, in addition to the traditional CVRFs identified since the 1950s, several new risk factors for atherosclerosis have been identified, including many diseases that increase systemic inflammation (such as gout, inflammatory bowel disease, autoimmune collagen vascular diseases, and psoriasis), some factors that occur during maternity (pre-eclampsia, delivering a child of low birth weight, preterm delivery) or child-bearing age (premature or surgical menopause) in females; factors that occur in early childhood (early-life trauma in young and middle-aged individuals with a history of AMI) and some lifestyle factors other than unbalanced diet and sedentary behavior, such as low socioeconomic status and air pollution.

Since 2003, with the pioneering works of Iacobellis et al., the presence of epicardial fat has also been increasingly proposed and better defined as a risk factor, not only for coronary atherosclerosis but also for atrial fibrillation and HF, in relation to its localization at the atrial or pericoronary level, respectively [153,154]. Adipocyte aggregates between the myocardium and the visceral layer of the epicardium have unique anatomic and functional characteristics, given their close proximity and interactions with the heart owing to shared circulation and the absence of muscle fascia separating the two organs. EAT can be clinically measured with cardiac imaging techniques that can help to predict and stratify cardiovascular risk [102].

Finally, to overcome the limitations of observational and cohort studies, genetics was used to better define the cause-and-effect relationships between CVRFs and CV events. Large-scale genome-wide association studies (GWASs) conducted over recent decades have identified numerous genetic variants associated with various cardiometabolic profiles and risk factors. From these discoveries came the so-called Mendelian randomization studies that use the genetic differences present in the study population as a “natural experiment” to improve inferences about the cause-and-effect relationship between hypothetical CVRFs and CV events derived from prior observational studies. Mendelian randomization studies have several advantages over RCTs, given that they are usually cheaper and faster to create, since they are derived from the existing, large-scale GWAS database.

### 5.2. Biomarkers

An ideal clinical biomarker should contain the following characteristics [155]: (1) clinical relevance, (2) sensitivity and specificity to treatment effects, (3) reliability, (4) practicality, and (5) simplicity. Based on these general principles, it is possible to profile the ideal biomarker for the atherosclerotic process (Table 3).

The atherosclerotic process involves multiple pathophysiological mechanisms. Inflammation of the vessel wall has long been recognized as central to the initiation of the atherosclerotic process, as well as to its progression and transition to stages of clinical instability. Not surprisingly, many biomarkers directly or indirectly related to the inflammatory process (Table 4) have been proposed as useful in monitoring atherosclerotic pathology, first in experimental models and then clinically. The endothelium can be considered the target organ of the pathophysiological processes that initiate atherosclerosis. A recent review pointed out the differences between an activated endothelium and a dysfunctional endothelium, the former being a very early stage of dysfunction [156]. Biomarkers of endothelial activation are endothelial adhesion molecules, cytokines, C-reactive protein, CD62E+/E-selectin activated endothelial microparticles, oxidation of LDL, asymmetric dimethylarginine, and endocan. Biomarkers of endothelial dysfunction are matrix metalloproteinases such as MMP-7, MMP-9, ANGPTL2, endoglin, annexin V, endothelial apoptotic microparticles, and serum homocysteine. The recent discovery of exosomes, both as diagnostic and therapeutic tools, has boosted research on their use. Exosomes that derive from ECs may be responsible for the changing phenotype of vascular smooth muscle cells [157,158].

### 5.3. Imaging Techniques

Since coronary angiography, in particular, is invasive, expensive, not widely available, and not a risk-free procedure, clinicians and researchers have focused on two alternative strategies: the first has been the aforementioned study of biomarkers and genetic variants that can predict future atherosclerosis-related clinical events, while the second has been the study of atherosclerosis in more accessible vascular districts, such as the carotid and lower extremity arteries, using noninvasive methods, mainly ultrasound.

A unique feature of imaging is the ability to study calcifications in the coronary arteries.

(i)Coronary artery calcium (CAC) has become a useful tool to detect and quantify calcified subclinical atherosclerotic burden. The most widely used method to quantify CAC is the Agatston method, which uses the product of the total calcium area and a quantized peak calcium density weighting factor defined by the calcification attenuation in Hounsfied units on non-contrast computed tomography [159]; in addition, CAC may be identified on scans scheduled for other reasons [160]. In the ITALUNG trial, CAC was assessed in baseline, low-dose computed tomography performed on 1364 participants aged 59–69 years and with a smoking history ≥ 20 pack-years in a lung cancer screening program with a follow-up of 22 years. CAC score was graded as absent, mild, moderate, and severe. In the study, moderate or severe CAC was significantly associated with CV mortality after adjustment for traditional CVRFs [161]. CAC progression may also be a marker of accelerated atherosclerosis, as shown in a recent study regarding ICI therapy in cancer [142]. CAC score is unreliable with statin or PCSK9 inhibitor treatment [162,163].(ii)Computed Tomography Angiography. The “actionable lump” concept underscores the importance of early detection and proactive monitoring of sublinical atherosclerosis when preventive interventions, such as conversion to a healthy lifestyle and early treatment of CVRFs, may limit the progression of the atherosclerotic process [164].

There is indeed a large body of publications in the literature examining the promise of imaging markers for early detection of subclinical CVD to ameliorate a primary prevention approach. Available data are in favor of CAC as a strong marker of ASCVD risk. On the other hand, the absence of CAC or zero CAC has an even greater value as a negative predictive tool, with a remarkably favorable prognosis in older adults with guideline recommendations to consider deferral of lipid-lowering therapies in subjects with zero CAC [165,166]. Another intriguing finding is breast arterial calcification (BAC), a form of medial artery calcification that can be detected in routine mammography, which has been proposed in recent years as a sex-specific imaging marker for the early detection of subclinical CVD. However, the pathophysiology of CAC, BAC, and calcium deposits at other cardiovascular sites (valve and aorta calcification) differs, as do their prognostic implications. For example, BAC is a form of medial artery calcification whose formation is regulated by the expression of osteogenic genes in a manner similar to bone tissue. Unlike the intimal calcifications expressed by CAC, BAC is formed independently of lipid deposition and macrophage activation, processes typical of classic atherosclerosis. However, these intimal calcifications are thought to contribute to endothelial dysfunction and reduced vascular compliance. Similarly, valvular calcifications are likely to result from mechanisms that combine classic atherosclerotic and more strictly osteogenic processes. Even if BAC, CAC, aortic calcification, and valvular calcification are correlated with each other, and all have been associated with CVD, we have to take into account the aforementioned heterogenous pathophysiologic processes involved in their origin, which could make a difference in their clinical utility as an early marker of atherosclerosis. In conclusion, we can state that BAC, as well as other areas of extracoronary calcifications, can be useful as a general cardiovascular risk marker, and in this sense, its identification should lead to a vigorous implementation of Life Simple 7, having a positive impact on both CV risk and oncology risk. However, at present, the absence of BAC (zero BAC) does not have the same negative predictive value as zero CAC and is not equivalent to low CV risk [167].

In a paper published in 2024, Parveen Grag et al. [168] reported that carotid plaque burden, CAC, and low ankleؘ–brachial pressure index were the only three tests that robustly predicted future atherosclerotic events in people of middle age or older. The authors conducted a critical review of measurements used to infer the presence of subclinical atherosclerosis in the major conduit vessels and focused on the predictive capability of these tests for future CV events (defined as stroke, myocardial infarction, and chronic ischemic limb disease) independent of the presence of conventional CVRFs. The authors preferred studies with >10,000 person-years of follow-up limited to carotid, coronary, aorta, and lower limb arteries and performed a meta-analysis of the results, reporting adjusted hazard ratios (HRs) with 95% confidence intervals. In the carotid artery (eight studies), the presence of plaques was independently associated with future strokes (HR 1.89, 95% CI 1.04–3.44) and cardiac events (seven studies, HR 1.77, 95% CI 1.19–2.62). Coronary calcifications (five studies) were found to be associated with acute coronary events (HR 1.54, 95% CI 1.07–2.07), while an increase in their severity, as expressed by the Agatston score, was associated with a significant increase in risk in thirteen studies. Seven studies showed that an ankle–brachial index (ABI) < 0.9 was associated with an increased risk of cardiovascular death (HR 2.01, 95% CI 1.43–2.81).

## 6. What Is New on the Horizon for Early Atherosclerosis Imaging?

### 6.1. Magnetic Resonance Imaging (MRI)

The ability of MRI to accurately detect the presence of lipid deposits, the sine qua non of atherosclerosis, in various vascular territories, has been known for more than 20 years [169]. Technological advances that will soon make more “portable” MRIs available suggest that this method may also be used in large epidemiologic studies [170]. More recently, the combination of MRI and positron emission tomography (MRI-PET) has proven to be a promising imaging modality for studying the inflammation that underlies the various stages of the atherosclerotic process, with the possibility of whole-body studies [171].

The use of near-infrared spectroscopy, which chemically characterizes plaque, in conjunction with various intravascular imaging techniques (most commonly, intravascular ultrasound or optical coherence tomography) allows accurate study of the composition and characteristics of atherosclerotic plaques, although this method requires an invasive approach [172].

### 6.2. Computed Tomography Angiography (CTA)

In addition to aspects more closely related to anatomy, coronary CTA (CCTA) techniques can and will provide increasingly sophisticated information about plaque characteristics and the degree of associated inflammation [173]. All this is very useful because coronary risk assessment is a multiparametric process that includes clinical factors, such as conventional and non-conventional CVRFs, and anatomical factors, such as degree of stenosis and global plaque burden, but also biological characteristics of plaque, among which cellular composition and degree of associated inflammation seem to be very relevant.

Beyond the capacity of identifying the extent, distribution, and characteristics of high-risk coronary plaques (low attenuation, napkin ring sign, positive remodeling, and patchy plaque calcification), CCTA has been shown to be able to explore the biological mechanisms underlying CAD progression and clinical events through the visualization of coronary perivascular adipose tissue (PVAT), a very promising technique that examines PVAT as a “telltale” sign of the presence and degree of inflammation at the level of the coronary wall [174]. The parameter that expresses the degree of vascular inflammation is the Fat Attenuation Index (FAI) score, which can be obtained from routine CCTAs and can provide an assessment of inflammation for each individual coronary segment, integrating these data with clinical and anatomical data in the construction of an increasingly personalized risk profile [175].

## 7. Limitations and Unresolved Questions

However, it should be recalled that there is currently no solid evidence to support specific therapeutic interventions in the presence of subclinical atherosclerosis. It is certainly very suggestive to think that aggressive interventions on CVRFs such as dyslipidemia or high inflammatory burden may be able to alter the natural history of preclinical atherosclerosis, but this needs to be demonstrated with ad hoc-designed clinical trials, which will not be easy to conduct because they require adequate sample sizes and long follow-ups. In addition, to date, studies of subclinical atherosclerosis have typically focused on middle-aged or older populations (range of 40 to 70 years). It is possible that the identification of subclinical atherosclerosis may be of even greater value in younger individuals, in whom it is likely that currently used tests may not be sensitive enough to identify the earliest atherosclerotic lesions. This, of course, also underscores the need to develop new diagnostic methods capable of identifying the earliest stages of the disease.

## Figures and Tables

**Figure 1 ijms-25-04232-f001:**
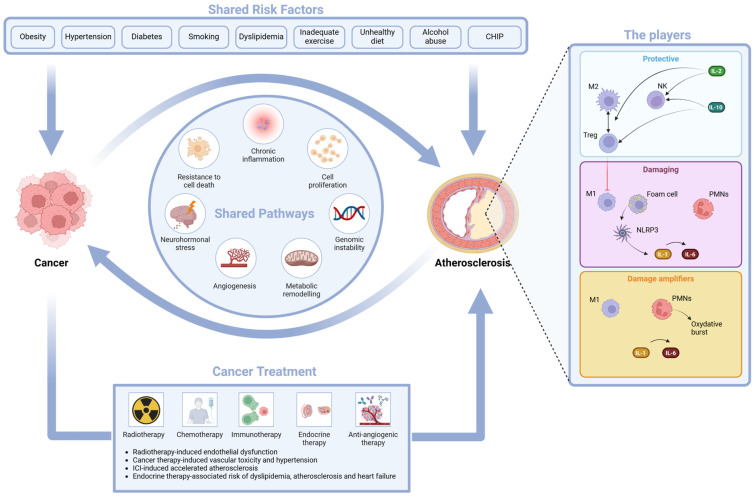
Central illustration of atherosclerosis and cancer: shared risk factors, shared pathways, cancer therapy-induced vasculotoxicity; the characters of the atherosclerotic play: the protective characters, the damaging characters, and the damage amplifiers.

**Table 1 ijms-25-04232-t001:** The chameleonic vascular and immune players of atherosclerosis with their “good”, “bad”, or “ugly” behaviors.

Players	Phenotypes	“Good” Behavior	“Bad” Behavior	“Ugly” Behavior
Endothelial cells (EC) secrete vasoactive substances (e.g., endothelin-1, nitric oxide, etc.) affecting vascular smooth muscle cells, platelets, and white blood cells. In atherosclerosis ECs are activated by trapped lipoproteins. Refs. [51,52,58,61]	Heterogeneous ECs to suit heterogeneous endothelium, capable of angiogenic and metabolic switch. EC may exhibit trained immunity.	Healthy ECs are excellent sensors of the hemodynamic forces of blood flow. ECs have a pivotal role in endothelial resilience, the ability to cope with many stressors or challenges (exposomes).	In the early stages of atherosclerosis, high levels of ox-LDL and remnants of tryglyceride-rich lipoproteins gain access to the subendothelial space, eliciting a danger signal that activates the NLRP3-inflammasome in innate immune cells and the inflammatory pathways, leading to endothelial dysfunction.	Inflammation begets inflammation, and this vicious circle leads to advanced stages of atherosclerosis.
Pericytes: perivascular cells derived from human pluripotent stem cells (HPSCs) and located around ECs. Refs. [62,63,64,65]	Due to their common origin from HPSCs, pericytes can differentiate into other cells of the mesenchymal lineage such as monocytes.	Support vascular stability by preventing matrix degradation; play a relevant role in differentiation, angiogenesis, regeneration, immunomodulation, and blood flow regulation.	Dysmetabolic-driven alteration of pericytes in diabetes contributes to plaque formation.	In advanced atherosclerosis, pericytes are involved in plaque neovascularization, inflammation, and vascular calcification processes.
Vascular smooth muscle cells exhibit a contractile phenotype in the healthy arterial wall. If stimulated by ECs through PDGF-BB and TNF-α, they can switch to a synthetic phenotype that increases the production of ECM, exosome, and proinflammatory cytokines. Refs. [66,67,68,69]	Various phenotypes with beneficial and detrimental role in atherogenesis.	VSMC stabilize fibrous cap in advanced atherosclerosis and produce ECM (fibroblast-like features).	Lipid-induced transformation in macrophage-like and foam cell-like phenotypes, exhibiting proinflammatory behavior and increasing vulnerability to plaque (macrophage-like features).	
Dendritic cells (DCs) bridge the innate and adaptive immune response involved in the scenario of evolving plaque. Refs. [66,70]	Preclinical studies: proatherogenic and anti-atherogenic function.	In Ldlr−/− mice fed with high-fat diet, autophagy disruption in DCs limits atherogenesis.	In humans, dendritic cell numbers are connected to vulnerability of atherosclerotic plaque.	
Monocytes in a homeostatic state populate blood, bone marrow, and spleen. Refs. [71,72]	Classical vs. non-classical monocytes. Monocytes may be rewired by metabolic stimuli (e.g., ox-LDL) to become a “trained” immune cell.	Non-classical monocytes are “on patrol” to maintain vascular endothelium. Recruited monocytes also have an impact on atherosclerosis regression.	Classical monocytes (CD14^+^ CD16− in humans, Ly6C^high^ in mice) recruited to atherosclerotic plaque exhibit phenotypic heterogeneity, differentiating into dendritic cells and macrophages.	In preclinical studies in mice, splenic classical monocytes (Ly6C^high^) increase plaque and its instability. In humans and mice, monocytosis is associated with increased severity of atherosclerosis.
Macrophages Refs. [73,74,75,76,77]	M1 macrophages, M2 macrophages. Macrophages may be upgraded in “trained” immune cells.	M2 macrophages clear lipids and secrete anti-inflammatory factors (e.g., Il-10 and collagen). Recruited monocyte-derived macrophages remove apoptotic cells (efferocytosis), eliciting the secretion of anti-inflammatory cytokines and hampering the progression of atherosclerotic plaque.	M1 macrophages favor the accumulation of intracellular lipids and increase the secretion of proinflammatory factors (e.g., TNF-α IL-1 β and IL-6). Macrophages may appear as foamy cells.	M1: when reprogrammed macrophages lose their efferocytic ability, apoptotic cells undergo post-apoptotic necrosis, releasing proinflammatory mediators and providing a boost to the progression of plaque.
T cells have a role in all stages of atherosclerosis. CD4+T cells are prevalent in mouse atherosclerotic plaque and exhibit a proinflammatory atherogenic phenotype. Refs. [78,79,80]	Atheroprotective phenotype (T reg) and proatherogenic phenotype (T helper 1).	T reg can silence inflammation through the elaboration of the immunomodulatory cytokine transforming growth factor beta and by secreting IL-10 (preclinical studies).	Proinflammatory phenotype (T helper 1 cells): activated T cells have a direct role in the arterial wall or help B cells in antibody production.	
B cells are classified into B1 cells (subdivided into B1a and B1b cells), mainly produced in the fetal liver, and B2 cells (subdivided in T1 and T2 marginal zone progenitor). Refs. [81,82,83]	B1 cells are atheroprotective in mice. When challenged by a high-fat/high-cholesterol diet, marginal zone B cells switch to an atheroprotective programme mediated by Atf3, Nr4a1, and Pdl1.	B1 cells exhibit atheroprotective behavior in mice due to the production of IgM antibodies that block the uptake of oxLDL by macrophages in atherosclerotic lesions.	B2 cells exhibit proatherogenic behavior through antibody responses that stimulate adaptive immunity.	
Neutrophils have a role in all stages of atherosclerosis Refs. [84,85,86]	Proatherogenic phenotype. Reparative phenotype.	Reparative phenotype exhibited during thrombotic events when neutrophils promote endothelial repair and angiogenesis (arterial healing).	Neutrophils secrete ROS, increasing the permeability of ECs and inducing NLRP3 inflammasome. Neutrophils attract monocytes and can activate macrophages via extrusion of their NETs.	NET formation that stimulates the NLRP3 inflammasome and produces IL-1β (preclinical study). In this scenario, NLRP3 inflammasome requires a second hit to be fully activated. Defective efferocytosis that leads to accumulation of DAMPs.

Legend: DAMPs, danger-associated molecular patterns; ECM, extracellular matrix; NET, neutrophil extracellular trap; NLRP3, nucleotide oligomerization domain-containing, leucine-rich repeat-containing protein3, and pyrin domain-containing protein 3; PDGF, platelet-derived growth factor; TNF-α, tumor necrosis factor-α; Atf3, activating transcription factor-3; Nr4a1, nuclear receptor subfamily 4 group a1; Pdl1, programmed death ligand-1.

**Table 3 ijms-25-04232-t003:** Ideal atherosclerosis biomarker.

Safe and easy to measure;
Reliable;
Sensitive and specific;
Capable of discriminating healthy patients from unhealthy patients;
Able to predict future cardiovascular events;
Should express early in the disease progression;
Can be applied to diagnosis, staging, and prognosis;
Cost efficient for follow-up;
Modifiable with treatment.

**Table 4 ijms-25-04232-t004:** Pro- and anti-inflammatory cells and mediators in atherosclerosis.

Experimental	
Biomarker	Atherosclerosis
IL-1 β	Anti IL-1 β-L: 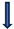 late atheroLOF IL-1 β: 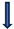 athero
IL-1 α	Anti IL-1 α: 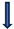 early atheroLOF IL-1 α: 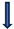 athero
IL-6/IL-6	LOF IL-6: 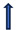 atherorIL-6: 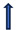 athero
IL-10	GOF IL-10: 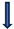 athero
T reg	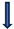 athero
**Clinical**	
**Biomarker**	**Atherosclerosis**
PMN	PMN 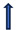 late lesions and atherothrombosis
IL-1 β	Anti IL-1β-L: 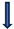 MACE
IL-6	Anti IL-6 L: 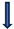 inflammation and thrombosis

IL: Interleukin; LOF loss of function; GOF: gain of function; rIL: reverse IL; MACE: major adverse cardiovascular events; 
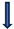
 decrease; 
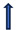
 increase. Modified from Matter MA et al. [103].

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
