# Peer review of "Atherosclerosis and the Bidirectional Relationship between Cancer and Cardiovascular Disease: From Bench to Bedside—Part 1"

_ijms, 2024, doi:10.3390/ijms25084232_

Round 1

Reviewer 1 Report

Comments and Suggestions for Authors

The paper ijms-2944449 is a comprehensive, very informative narrative review of the bidirectional relation between malignancies and atherosclerosis. The text is well structured and generally skillfully written.

I have a few minor comments:

Lines 186-199 and Table 2. In the context of clonal hematopoiesis of indeterminate potential (CHIP), the authors might wish to cite a subgroup analysis of the CANTOS trial (JAMA Cardiol 2022; 7: 521–528), which found that subjects with CHIP due to TET-2 loss of function variants responded to canakinumab much better than patients without CHIP.

Line 250: growth differentiation factor-15 (GDF-15)

Line 465: Please, state explicitly that life-style and pharmacologic interventions are meant, not catheter interventions.

What is new on the horizon of early atherosclerosis imaging? , lines 511-525. The authors might wish to add a few lines on the advances of CT techniques in characterizing soft-tissue plaque.

Technical comments:

Line 75: atherosclerosis (no capital letter needed)

References 87, 88, 109, 38 (in line 269) in the text are in superscript.

Ref. 101 in the text is bolded.

Line 230:  cardio-oncology

4.1. The shared risk factors, lines 233-235. A full sentence would be welcome.

Line 267: “As a of facts…” Please, correct.

Conflicts of interest: ..the results”. (Please, delete.)

Ref 95 in the References is written as 9595.

Comments on the Quality of English Language

-

Reviewer 2 Report

Comments and Suggestions for Authors

In this review Gallucci et al. analyzed the recent translational data on atherogenic cardiovascular and oncological diseases. The authors summarized the main vascular and immune players of atherosclerosis as well as the effects of the anti-inflammatory agents on the treatment of atherosclerotic diseases. The authors also describe cross-talk between the risk factors and pathways, the protective and the damaging characters in atherosclerosis and cancer. The review is well written and contributes to our knowledge on crucial cardiovascular and oncologic pathways and their interplay.

Minor comments:

1.    Please add a list of abbreviations.

2.    The summary is missing. What are the main conclusions and future perspectives?
